# SLAMuZero: Plan and Learn to Map for Joint SLAM and Navigation

**Primary Keywords:** *(1) Applications; (2) Learning*

## Abstract

MuZero has demonstrated remarkable performance in board and video games where Monte Carlo tree search (MCTS) method is utilized to learn and adapt to different game environments. This paper leverages the strength of MuZero to enhance agents' planning capability for joint active simultaneous localization and mapping (SLAM) and navigation tasks, which require an agent to navigate an unknown environment while simultaneously constructing a map and localizing itself. We propose SLAMuZero, a novel approach for joint SLAM and navigation, which employs a search process that uses an explicit encoder-decoder architecture for mapping, followed by a prediction function to evaluate policy and value based on the generated map. SLAMuZero outperforms the state-of-the-art baseline and significantly reduces training time, underscoring the efficiency of our approach. Additionally, we develop a new open source library for implementing SLAMuZero, which is a flexible and modular toolkit for researchers and practitioners (https://anonymous.4open.science/r/Anomalous-F517/).

## 1 Introduction

Imagine you find yourself in a new city and need to reach a specific destination. How would you like to navigate your way out? Like most people, you would likely rely on a map. You also need to first identify "your position" on the map. This scenario draws an analogy to robotic navigation, where a robotic is tasked with exploring a new room (Figure 1a) while building a map on the top of it (Figure 1b).

Simultaneous Localization and Mapping (SLAM) is a widely employed technique in the fields of robotics and computer vision. It allows robots and devices to simultaneously create a map of their environment while determining their own position within that environment. SLAM enables a robot or sensor to navigate and understand an unknown or partially unknown space by both building a map of the surroundings and continuously updating its own position relative to that map, which is crucial for autonomous vehicles, drones, and other robotic systems that need to operate in unfamiliar or changing environments (Placed et al. 2023).

The rapid advancement in deep reinforcement learning (DRL), the widespread availability of user-friendly simulators (Beattie et al. 2016; Savva et al. 2019), and the accessibility of open-source datasets from cameras and sensors (Xia et al. 2018; Chang et al. 2017; Ramakrishnan et al. 2021)

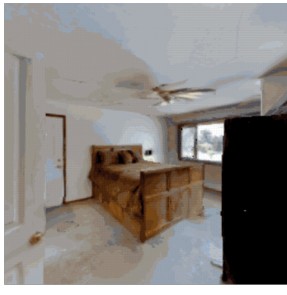 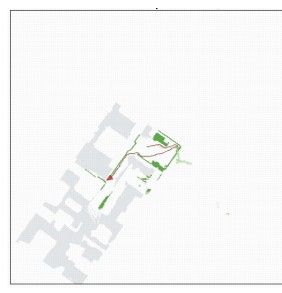

|  (a) Observation | (b) Map and location |

Figure 1: Motivating example of SLAM and navigation

have stimulated a surge of research interest in active SLAM (Mirowski et al. 2016; Chen and Gupta 2019; Chaplot et al. 2020a,b). This paper introduces a novel learning framework referred as SLAMuZero, designed to address the joint challenges of SLAM and navigation. Within this framework, we leverage the power of Muzero, a highly regarded planning module in DRL known for its remarkable performance in handling complex action spaces across a variety of game environments (Schrittwieser et al. 2020; Hubert et al. 2021; Antonoglou et al. 2021). Our contributions include:

1. We introduce SLAMuZero for joint SLAM and navigation. SLAMuZero extends the search process in MuZero by introducing a SLAM module that decodes the predicted map from the hidden state for path planning. SLAMuZero outperforms the state-of-the-art baseline and significantly reduces training time.

2. We provide a modular and extendable library for implementing SLAMuZero. This library provides a user-friendly toolkit for deploying MCTS-based algorithms and allows users to readily adapt MuZero and its variants to existing RL frameworks.

The rest of the paper is organized as follows: Sec. 2 discusses related work. Sec. 3 presents preliminaries regarding MuZero. Sec. 4 introduces our proposed SLAMuZero. Sec. 5 introduces the solution approach. Sec. 6 presents numerical results and Sec. 7 concludes.

## 2 Related Work

There has been a growing trend applying deep learning and reinforcement learning methods to SLAM (Placed et al.

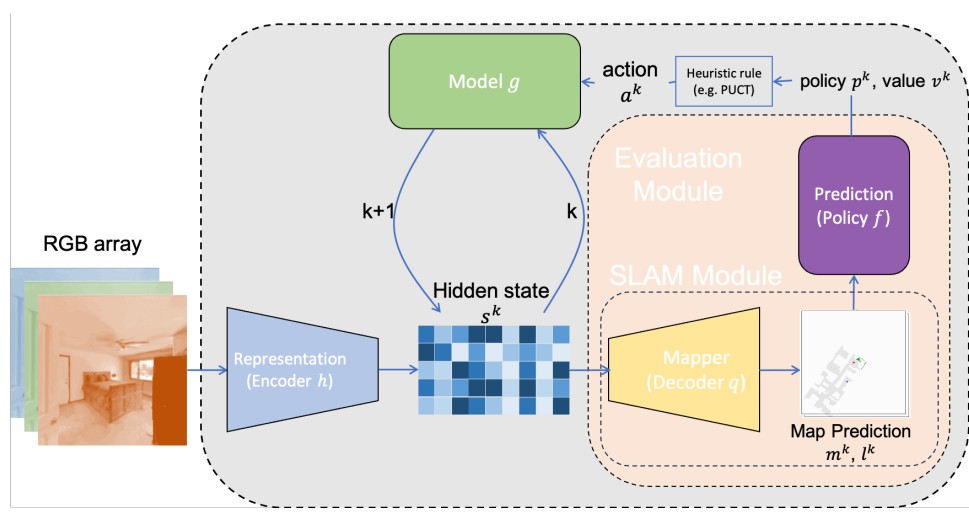

Figure 2: SLAMuZero architecture

2023). An end-to-end learning scheme is adopted by (Chen and Gupta 2019) for active SLAM conducted on the Habitat platform (Savva et al. 2019). Chaplot et al. 2020b uses a hierarchical learning scheme to achieve better performance for navigation tasks across different indoor environments. Oh and Cavallaro 2019 proposed a learning method that predicts the future camera view from a current state-action pair for exploration. Gottipati et al. 2019 leverages convolutional neural network for perception and model-free deep reinforcement learning for acting to disambiguate the agent pose within a reference map. Placed and Castellanos 2020 uses laser measurements as inputs to D3QN and guided the training with the reward to solve active SLAM. However, the current DRL methods for active SLAM lack the necessary planning abilities required for agents to navigate unknown environments effectively. To address this issue, our SLA-MuZero integrates a SLAM module with MuZero, resulting in enhanced planning capabilities and learning efficiency.

## 3  Preliminary

### MuZero

MuZero (Schrittwieser et al. 2020) is designed for RL tasks and is capable of learning to master complex environments and games without any prior knowledge of their rules. At the core of MuZero is Monte-Carlo Tree Search (MCTS) (Schrittwieser et al. 2020; Browne et al. 2012), a technique that has demonstrated superhuman performance across various game scenarios. MuZero is structured into three phases. In the planning phase, MuZero first encodes past observations into a hidden state. Subsequently, it executes MCTS, where the process initiates at the root node. At each encountered leaf node, MuZero employs a prediction function to evaluate the hidden state and generate priors for candidate actions (child nodes), along with computing the value associated with the node. At each non-leaf node, MuZero selects a child node recursively, continuing this process until it reaches a leaf node. Concurrently, it updates the hidden state in a recurrent manner with a dynamics function, which takes the previous hidden state and selected actions as input.

In the acting phase, an action is sampled from a distribution proportional to the visit count of the root node's children. In the training phase, for the evaluation function, the policy is aligned with the visit count distribution, while the value is trained to approximate the n-step bootstrapping return. For the dynamic function, the model is trained to minimize the difference between the predicted and the observed reward.

## 4  SLAMuZero

In this section, we introduce SLAMuZero, which consists of representation, SLAM, prediction and dynamics modules. The workflow of SLAMuZero is illustrated in Figure 3. In our SLAMuZero, the representation module $h$ and SLAM module $q$ constitute the Encoder-Decoder architecture, which is a mapping between sensor output and the estimated environment. The representation module encodes the raw observations into a hidden state. The model is unrolled for $K$ steps, where the hidden state is recurrently updated through dynamics module $g$. At each hypothetical step $k$, the hidden state is decoded by the SLAM module to generate the map and location as $m_t^k, l_t^k$. The policy and value are computed from the internal map and location $m_t^k, l_t^k$ by the prediction module, $p_t^k, v_t^k = f(m_t^k, l_t^k)$, which guides the search tree expansion. The search is performed at each timestep $t$. The next action $a_{t+1}$ is selected according to the visit count for each action from the root node of search tree.

We now introduce each module and how four modules are jointly trained in SLAMuZero (Figure 2).

- **Representation**: The hidden state is the output of the representation module $h$, which takes raw observations as input, such as an RGB image of the room to be explored. In this work, the module takes images as input, which have a resolution of 256x256 and consist of three channels. These three channels are rescaled to the range [0,1]. Given the large spatial resolution of the RGB observations, the representation module employs a sequence of convolutions with a stride of 2 to reduce the spatial resolution. The output is a hidden state with a resolution of 16x16 and 64 channels.

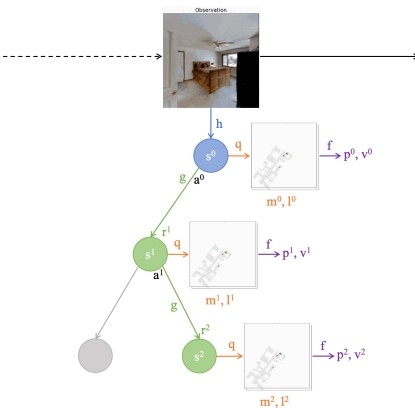

Figure 3: SLAMuZero workflow.

- **SLAM**: The map $m^k$ and location $l^k$ in the map $m^k$ are generated by the SLAM module. The module takes the hidden state as input and performs deconvolution to generate a decoded state with the same resolution as the ground truth map. It also reduces the number of channels to 2, with one representing the predicted map and the other representing the explored area. This decoded state is then passed into an estimator, which outputs the location using a combination of convolutions and fully connected layers.

- **Prediction**: For each pair of map $m^k$ and location $l^k$, the policy $p^k$ and value $v^k$ are output by the prediction module $f$. In the prediction module, the policy head and value head maintain the spatial resolution while altering the number of channels, followed by linear layers to map the size of the output to the number of actions and the value support size, respectively.

- **Dynamics**: Given a hidden state $s^{k-1}$ and a candidate action $a^{k-1}$, the dynamics module $g$ produces an immediate reward $r^k$ and a new hidden state $s^k = g(s^{k-1}, a^{k-1})$. The module initially encodes the action into a single panel with the same resolution as the hidden state, rescaled to the range [0,1]. This panel is then appended to the hidden state and passed through 8 Residual Blocks to generate the next hidden state with the same shape. To enhance the stability of the process, the hidden state is rescaled to fall within the range [0, 1] at each step. Additionally, the loss is multiplied by $\frac{1}{K}$ to ensure that the unroll step size has no impact on the magnitude of the total gradient.

## 5 Solution Approach

**Learning algorithm**

The algorithm for SLAMuZero is summarized in Alg. 1. For one sub trajectory sampled from replay buffer, for each timestep $t$, the representation module $h$ generates the initial hidden state $s_t^0$ from the past observations $o_{1:t}$ from the sampled trajectory. The model is subsequently unrolled recurrently for $K$ steps. At each step $k$, five pairs of quantities are collected to calculate the loss, namely the predicted policy $p_t^k$ with the action probability from the root node $\pi_{t+k}$, the preidcted value $v_t^k$ with the n-step bootstrapping value

$z_{t+k}$, the predicted reward $r_t^k$ with the actual reward received $u_{t+k}$, the predicted map $m_t^k$ with the ground truth map $d_{t+k}$, the predicted location $l_t^k$ with the actual location $e_{t+k}$. Then, the dynamics module $g$ receives the hidden state $s_t^{k-1}$ from the previous step and the real action $a_{t+k}$ as inputs to generate the next hidden state $s_t^k$. The model employs a loss function to minimize the discrepancies between predicted and actual rewards, values, and policies, as well as the discrepancies between estimated and actual maps and locations, commonly referred to as the "SLAM error". This loss function is defined as follows:

$$l_t(\theta) = \sum_{k=0}^{K} l^r(u_{t+k}, r_t^k) + l^v(z_{t+k}, v_t^k) + l^p(\pi_{t+k}, p_t^k)$$
$$+ l^{map}(d_{t+k}, m_t^k) + l^{pose}(e_{t+k}, l_t^k) + c||\theta||^2 \quad (1)$$

where, $l^r, l^v, l^p, l^{map}, l^{pose}$ are errors for reward, value, policy, map, location, respectively. $c||\theta||^2$ is a regularized term capturing the learning process parameterized by $\theta$.

---

**Algorithm 1: SLAMuZero**

1: **for** $t \leftarrow 1...N$ **do**
2:     $\pi_t \leftarrow$ planning($h(o_{1:t})$)     ▷ Sample action $a_t$ from root node's children visit count
3:     $a_t \sim \pi_t$.
4:     Execute $a_t$ in environment to collect $u_t$, ground truth $d_t, e_t$
5:     **for** $k \leftarrow 1...K$ **do**
6:         $m_t^k, l_t^k = q(s_t^k, \theta)$
7:         $p_t^k, v_t^k = f(m_t^k, l_t^k, \theta)$
8:         $a_t^k \sim p_t^k$
9:         $r_t^k, s_t^{k+1} = g(s_t^k, a_t^k, \theta)$
10:     **end for**
11:     $z_t = $ bootstrap($\{u\}_{1:t}$)
12:     Obtain loss according to Equ. 1
13:     $\theta \leftarrow$ optimise($l, \theta$)
14: **end for**

---

**Open source library**

To implement and evaluate the proposed SLAMuZero, we propose an open-source library tailored for customizing MuZero algorithms. This library is seamlessly integrated with the Habitat platform, which offers a realistic and controlled setting for testing, to perform visual based robot navigation tasks (Figure 4). The data source is detailed in Sec. 6.

We now introduce the design of the open-source library to facilitate the implementation of our proposed SLAMuZero. Despite the existence of numerous RL frameworks, such as stable-baselines (Raffin et al. 2021), rllib (Liang et al. 2018), and ElegantRL (Liu et al. 2021), the implementation of MCTS-based algorithms and their adaptation to specific tasks remains a challenge. The widely used EfficientZero (Ye et al. 2021), LightZero (Niu et al. 2023) suffer from difficulties in extending the algorithm to accommodate customized loss designs, customized neural network designs, and easy adaptation to different frameworks or environments. To address these issues, we develop a library

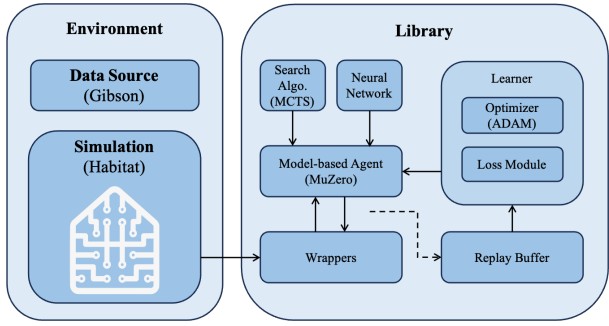

Figure 4: An overview of system architecture

for MCTS-based deep reinforcement learning as follows: A primary interface is provided to specify the search algorithm, network designs, as well as the learner, where loss and optimizer can be easily customized. The library employs mctx (Danihelka et al. 2021) as the backend for performing the search process, which can be fully compiled just-in-time and run in parallel, ensuring an efficient process. Designed for flexibility and extensibility, our library simplifies the adaptation of MCTS-based RL to prevailing RL frameworks. It empowers users to innovate with advanced neural networks, experiment with novel loss functions, and test in customized environments. More details can be found in (https://anonymous.4open.science/r/Anomalous-F517/).

## 6    Experiment

In this section, we conduct numerical experiments and make a comparison of our method and baselines.

### Experimental set-up

The exploration task involves the agent navigating in an unfamiliar environment with the goal of exploring as much space as possible (see Figure. 1). The agent receives observations in the form of RGB images from the visual sensor and pose information from the motion sensor. To guide the agent in the environment, we utilize an analytical path planner (Sethian 1996), which takes the agent's pose, the map, and the goal as inputs to calculate the robot's action sequence. The reward in this task is defined as the exploration ratio of the entire room, and the action is defined as the location on the predicted map.

The evaluation metric is the percentage of area explored in the scene (Cov), which is the ratio of coverage to the maximum possible coverage in the corresponding scene. The parameters for the Gibson scene are the same as those used in Neural SLAM (Chaplot et al. 2020a). Other parameters are listed in Table 1.

Table 1: The parameters for experiments

| Para | Val | Para | Val |
|------|-----|------|-----|
| Episodes | 30 | Vision Range | 64 |
| Frame Width | 256 | Simulations | 30 |
| Frame Height | 256 | Trajectories | 4 |
| Camera Height | 1.25 | N bootstrapping | 10 |
| Sample per Trajectory | 16 | K steps | 5 |

The data utilized in our experiments are Gibson (Xia et al.

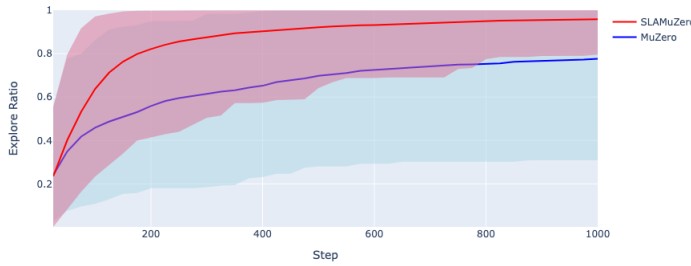

Figure 5: Coverage ratio per step on Gibson Val set

2018). The Gibson dataset offers large-scale, photorealistic environments for testing autonomous agents. It encompasses a variety of indoor environments, making it a suitable dataset for evaluating models across a wide range of scenarios. The task datasets can be found on the Habitat repository (Savva et al. 2019).

## Results

The proposed SLAMuZero is trained on the Gibson dataset with a maximum of 1 million frames in contrast to 10 million frames for other baselines. This indicates a significant improvement on training efficiency of our method. The result in Table. 2 is the averaged coverage ratio (Cov) over 280 episodes for 14 different unseen scenes from Gibson Val set. SLAMuZero achieves 0.961 final explore ratio in a comparison to 0.948 from the state-of-the-art baseline. To better understand the performance of SLAMuZero, we also add the result of Muzero without any SLAM module. Figure. 5 plots the explore ratio (Cov) of SLAMuZero and MuZero per step averaged over all Gibson Val set. The advantage of the SLAM module is akin to a shift in perspective, allowing control to take place on a map, while MuZero's advantage lies in planning with a leanred model.

Table 2: Comparison of our method and baselines

| Method | Cov |
|--------|-----|
| RL + 3LConv (Savva et al. 2019) | 0.737 |
| RL + Res18 (Chaplot et al. 2020a) | 0.747 |
| RL + Res18 + AuxDepth (Mirowski et al. 2016) | 0.779 |
| RL + Res18 + ProjDepth (Chen and Gupta 2019) | 0.789 |
| Active Neural SLAM (Chaplot et al. 2020a) | 0.948 |
| MuZero | 0.803 |
| **SLAMuZero** | **0.961** |

## 7    Conclusion

This paper has introduced the SLAMuZero framework, which integrates Simultaneous Localization and Mapping (SLAM) with the tree-search based MuZero. By combining these two robust techniques, we have demonstrated advancements in joint active SLAM and navigation, enhancing an agent's planning capabilities and enabling efficient navigation in unknown environments while concurrently constructing precise maps and accurately localizing the agent. Moreover, we have presented an open-source library designed to facilitate the implementation and experimentation of SLAMuZero.

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
