# OpenReview forum: "SLAMuZero: Plan and learn to Map for Joint SLAM and Navigation"
_icaps-conference.org/ICAPS/2024/Conference — ICAPS 2024_

### Official Review · Reviewer_t36M · 2024-01-18

**Significance And Importance:** 2
**Soundness:** 3
**Novelty:** 3
**Clarity:** 3
**Overall Evaluation:** 2
**Confidence:** 4

**Weaknesses:**

0: Minor weaknesses requiring some work to be addressed for the paper to be accepted.

**Contributions Of The Paper:**

This paper presents the SLAMuZero framework which combines the planning method (Monte-Caro Tree Search) with neural network structures for robotics SLAM and navigations. The authors have provided their source code and opensourced their contributions.

**Ethical Considerations:**

(1) Not Applicable: The paper does not have any ethical considerations to address

**Nomination For Best Paper:**

No

**Questions For Authors:**

1. Currently, SLAMuZero receives observations of RGB images. In more practical robotics applications, robots are often equipped with a wide range of sensors. Therefore, can SLAMuZero be easily adapted to multimodal inputs?

2. In real applications, is the dynamic module $g$ easy to obtain? Given $g$ takes a hidden state and an actual robot action as inputs, and outputs a new hidden state and an instant reward.

**Reproducibility:**

5: Code and domains (whichever apply) are already publicly available

**Strengths Of The Paper:**

1. The combination of "MCTS + Neural networks" has succeeded in many fields especially in games. In robotics, applying this architecture to SLAM is a reasonable and logical progress;

2. I appreciate that the authors provided their source code and opensourced their contributions, it makes other researchers easy to follow or use this work in robotics applications;

3. Overall, this paper is easy to understand and the experiment valid their approach compared with some recent learning approaches.

**Weaknesses Of The Paper:**

The authors used the Gibson dataset to train and valid their approach with other learning approaches. It would be better if the authors could test their SLAMuZero with different parameters. But considering this is a short paper, this should be a minor concern.

Some small issues:
   - please be careful with the "extra lines" in Figures 2 and 3;
   - the illustrations of decoded states in figures 2 and 3 are too small. For example, in figure 3, it's difficult to see the difference of robot's location $l^k$ along different trajectories;
   - In line 149, "the map $m^k$ and location $l^k$ in the map $m^k$" seems a bit strange, do you mean "the map $m^k$ and location $l^k$ in the decoded state"?

---

> ### Author Rebuttal · Authors · 2024-01-28
>
> Thanks for positive feedback. We will correct typos in the final manuscript.
>
> We answer questions as follows:
>
> R-Q1: Our method currently takes raw RGB and poses information from the motion sensor as input. Meanwhile, it is able to take multimodal input by adding complexity to the representation module. Nguyen et al. 2020 "Autonomous navigation in complex environments with deep multimodal fusion network.", Driess et al. "Palm-e: An embodied multimodal language model."
>
> R-Q2: To obtain a dynamic module that captures full dynamics of the entire environment is challenging, however, as Weirui et al 2021 and Schrittwieser et al 2020 pointed out, it is possible to train a dynamic module that only models the aspects that are important to the agent’s decision-making process.

---

### Official Review · Reviewer_1ozr · 2024-01-22

**Significance And Importance:** 2
**Soundness:** 3
**Novelty:** 2
**Clarity:** 3
**Overall Evaluation:** 1
**Confidence:** 4

**Weaknesses:**

0: Minor weaknesses requiring some work to be addressed for the paper to be accepted.

**Contributions Of The Paper:**

The paper suggests to combine MuZero (a technique for planning and learning to act) with SLAM (localization and mapping) in a rather natural manner to tackle joint SLAM and navigation in previously unknown environments. The idea seems very reasonable and the experiments conducted support the case. Another contribution is an implementation of an open-source library that enables to run SLAMuZero and extend it.

This paper presents a logical development for tackling joint SLAM and navigation by a new integration of known computational modules, which is not highly original but is still relevant and practical, and thus in my view is worthy of publication (subject to some extra work that is needed in the experimental part --  see "Weaknesses Of The Paper").

**Ethical Considerations:**

(1) Not Applicable: The paper does not have any ethical considerations to address

**Nomination For Best Paper:**

No

**Questions For Authors:**

1. Can you explain how the ground truth map and pose (d_t and e_t) are obtained in a physical environment (i.e., a non-simulated one)?
2. Please explain how you used the analytical path planner that is described in the "Experimental set-up". Where does it fit in the experiments?

**Reproducibility:**

5: Code and domains (whichever apply) are already publicly available

**Strengths Of The Paper:**

1. A reasonable idea to use the highly performant MuZero together with a SLAM module in order to solve the joint SLAM and navigation task.
2. Clear presentation of the idea, algorithm, high-level code implementation and experimental technical details.
3. Decent proof of concept (the experiments).
4. Extendable open-source implementation.

**Weaknesses Of The Paper:**

The only problem I have with the paper is related to the experiments and the experimental analysis, which are not sufficient in my view:
1. I would like to see results for another dataset, to give the results more credibility.
2. Similar experiments are required for a different evaluation metric. Specifically testing computation time on the fly (while the robot is acting) is interesting, as it might be that this approach is much more computationally demanding relative to other existing approaches.
3. Further statistical results (in addition to the empirical averages shown) are required in the presentation.
4. Analysis of the learning process is also relevant, in particular to examine the stability of the learning convergence.

It is perfectly fine to add all the above in an appendix.

---

> ### Author Rebuttal · Authors · 2024-01-28
>
> Thanks for your valuable feedback. We answer questions as follows:
>
> R-Q1: Good point. Our work utilizes the Habitat simulation environment in which the map is obtained by sampling navigable points from the environment and pose is obtained by querying the agent state. To obtain the accurate ground truth map and pose in a physical environment, various sensors may be used. The map generation and pose estimation can be obtained by LiDAR and IMU. And for smaller or controlled environments, manual measurements can be taken.
>
> R-Q2:  The analytical path planner is used for computing the actual movement (forward, left, right) using the Fast Marching Method (Sethian 1996), given the starting (agent position) and terminal (agent action) positions in a map. The agent’s actions are defined as positions of its predicted map. Upon selecting a terminal position based on the predicted map, the agent then waits for the robot to perform the movement as dictated by the analytical path planner for a predefined number of steps. In our experiments, the agent selects a new position every 25 steps.

---

### Official Review · Reviewer_uqBZ · 2024-01-23

**Significance And Importance:** 1
**Soundness:** 2
**Novelty:** 2
**Clarity:** 1
**Confidence:** 4

**Weaknesses:**

-1: Major weaknesses requiring significant work to be addressed for the paper to be accepted.

**Contributions Of The Paper:**

This paper presents SLAMuZero, a framework that integrates MuZero (Schrittwieser et al. 2020) with a SLAM module. The architecture of the approach uses an encoder to represent the hidden state and a decoder to create a map and predict the location. It then produces a policy and selects an action that will improve the map and location predictions.

**Ethical Considerations:**

(2) Poor: The paper fails to address crucial ethical considerations

**Nomination For Best Paper:**

No

**Overall Evaluation:**

-1: (weak reject)

**Questions For Authors:**

1. How would a robot using your method have the resolution of the ground truth map available to it? If the environment is unknown then it would not have this information, but you provide it to your system (line 152-153).
2. Why is the goal of the method to explore as much space as possible? (line 241) Should it be to explore as much as possible within a given time limit or step count?

**Reproducibility:**

5: Code and domains (whichever apply) are already publicly available

**Strengths Of The Paper:**

The paper presents a novel approach that combines MCTS with SLAM.

**Weaknesses Of The Paper:**

The paper has some weaknesses and the work seems premature for publication. (The following comments have been edited to reflect the rebuttal from the authors)
1. One issue is that the evaluation is not very strong. The comparisons in Table 2 are all just taken directly from Chaplot et al. 2020a (except for MuZero and SLAMuZero which the authors ran but with a different set of parameters and scenes). Additionally, the results only show metrics from the Gibson validation set, not on generalization to the MP3D environments.
2. The paper also has only one mention of the path planner they use, it would be useful to have some additional details.
3. There is no indication of what kind of hardware was used to run the experiment and whether that hardware setup matches the baseline comparisons.
4. Also, the results do not mention how long it took to train their method. If the paper's method only improves the coverage ratio by 0.013 but takes 10 times longer to train, it may not be as much of an advancement. The paper does say that it improves on training efficiency (1 million frames vs 10 million for other methods), but that still does not address the training time.
5. Another issue is that the paper should be careful about conflating navigation and exploration. Exploration is usually considered one aspect of navigation, but the most common navigation task is PointGoal. A system trained for exploration may not be successful at PointGoal navigation.

Additionally, there are many issues with the writing and organization that make the paper difficult to follow.
For example, there are multiple typos: "a robotic", "Muzero" "Table. 2", "learned". There are also run-on sentences and inconsistent reference styles ("adopted by (Chen and Gupta 2019)" vs "Oh and Cavallaro 2019 proposed"). The term "sub trajectory" is used without explanation. The explanation of MuZero is not clear, especially in the second half starting at line 108. Figure 3 is referenced before Figure 2, even though Figure 2 comes first in the paper. It is unclear what Figure 3 represents. Figure 4's reference in the text does not relate to the content of Figure 4. Gibson scene is mentioned on line 254 before it is described in the next paragraph.

---

> ### Author Rebuttal · Authors · 2024-01-28
>
> We would like to first clarify that (1) In the active neural SLAM method, the ground truth map is utilized in the training phase and is not accessible in the evaluation phase. As for our method, ground truth is for validation purposes, and is not accessible as well in the evaluation phase. The parameter settings and scenes are the same as Chaplot et al. 2020. (2) The performance metrics are conducted on the test data. (3) As for hardware, 2 NVIDIA T4 GPUs are used, with 8 vCPU (4 cores, 52GB memory). We will add this to the final manuscript. (4) The training efficiency is provided in the old manuscript. (See section 6). We pointed out in our paper that the result is obtained with 10% total training frames in a comparison to previous methods. (5) We will change the term “navigation” to “exploration” and correct our typos in the final manuscript.
>
> We then answer questions as follows:
>
> R-Q1: The resolution of the map is predefined and holds still for training and evaluation, and thus is invariant regardless of the scenes.
>
> R-Q2: Yes, we agree with the reviewer that the goal is to explore as much as possible within a time limit. We would like to clarify that we do set up the time limit. The reward is given as “explore as much as possible”, while at the same time 1) we use the same setup as the baseline, which set a step limit of 1000 steps for each scene 2) we apply discount to each timestamp, thus a higher total reward is obtained by exploring as much and as quickly as possible.

---

### Meta-Review · Area_Chair_ihEW · 2024-02-06

**Recommendation:** Accept (Oral)
**Confidence:** 2

**Metareview:**

The paper studies a problem of navigating in an unknown environment and suggests a combination of Monte-Carlo Tress Search (MCTS) with several learnable simultaneous-localization-and-mapping (SLAM) modules to solve the problem.

The strength of this (short) paper is that the combination of MCTS and SLAM is novel and is showed to outperform several baselines. On the other side, the empirical evaluation does not seem to be comprehensive. Including more competitors and, probably, evaluating on different problem setups (as currently only the exploration scenario is addressed) is worthwhile.

Another concern is that it is questionable whether the paper fits the ICAPS scope. Indeed, ICAPS community is interested in robotic problems as well as in machine learning, but the 'planning' component should still be clearly articulated. Here the latter is present in the form of Monte-Carlo Tree Search, a well-known method which is not significantly modified. Indeed, the main novelty lies outside the 'planning' domain (in integrating MCTS with the other task-specific methods). Consequently whether this contribution is significant from the Planning point of view is questionable. I wonder whether the inclusion of this paper into the ICAPS'24 program is a win-win for both the authors and the community. It may be the case that this paper would get more attention and impact if presented at ML/Robotics conference and, at the same time, it is not clear what 'planning lessons' the ICAPS community can learn from the paper.

Finally, as the study involves training deep neural networks the authors are encouraged to pay more attention to privacy and surveillance concerns, biases and assumptions built into the model and training data, and the environmental impact of the computation needed to train deep learning methods.

**Ethical Considerations:**

(1) Not Applicable: The paper does not have any ethical considerations to address